# Interdiffusion in Refractory Metal System with a BCC Lattice: Ti/TiZrHfNbTaMo

**DOI:** 10.3390/e25030490

**Published:** 2023-03-12

**Authors:** Mikhail I. Razumovsky, Boris S. Bokstein, Alexey O. Rodin, Alexandra V. Khvan

**Affiliations:** 1Department of Physical Chemistry, National University of Science and Technology (NUST) MISIS, 119049 Moscow, Russia; 2Thermochemistry of Materials SRC, National University of Science and Technology (NUST) MISIS, 119049 Moscow, Russia

**Keywords:** interdiffusion, high entropy alloy, refractory metals, BCC lattice, interdiffusion parameters, phase diagram, Hall’s method

## Abstract

Interdiffusion of the elements in a diffusion pair consisting of Ti and an equiatomic high-entropy alloy (HEA) TiZrHfNbTaMo in the temperature range of 1473–1673 K has been studied. A calculated results phase diagram of the alloy by Thermo-Calc 2021-B software as used to determine the temperature stability range of the β-phase in the alloy. Ti–HEA diffusion pairs were obtained by low = temperature welding and then diffusion annealing was carried out at temperatures of 1473, 1573, and 1673 K during 12, 9, and 6 h, respectively. The interdiffusion zone was profiled using electron probe microanalysis (EPMA). The diffusion parameters of the HEA’s elements were obtained using Hall’s method. An experimental results discussion is given.

## 1. Introduction

Usually, the composition of functional materials consists of the main component, which provides the main requirements for the material, and the alloying additions to improve the properties. A typical example of such materials is Ni-based heat-resistant alloys (HRA), where Ni is chosen as the main component due to some characteristics of this metal that favor operation in a wide temperature range (from room temperature to high operating temperatures) under load [1,2,3]. The operating temperatures of nickel HRA’s are limited from above by the melting point T_m_ ≈ 1725 K and do not exceed 0.75 T_m_ (T_m_ ≤ 1400 K). However, aerospace technology requires the development of HRAs for long-term exploitation at operating temperatures about 1600–1700 K. To work in such conditions, materials which have higher melting temperatures, i.e., refractory metals and compounds, are needed.

Currently, producing metallic HRAs based on many refractory metals, the so-called high-entropy alloys (HEAs), is being actively investigated [4,5,6,7]. Such alloys contain more than 5 basic elements with a concentration of each from 5 to 35 at.% and in addition to the high melting point, they are characterized by a large configuration entropy S_conf_.

Since the increase in entropy is associated with the increased disorder in the system, the suggestion was made that along with other factors, this may contribute to limiting effective mass transfer [4].

Today there is very little data available on the dependence of the interdiffusion coefficients on the composition of the alloy in binary systems of refractory metals, even less for multicomponent alloys with a BCC lattice. There are much more studies on diffusion in multicomponent HEAs with an FCC lattice than on those with a BCC lattice. The objective of this study was to obtain data for multicomponent systems with a BCC lattice based on many refractory metals.

Along with the large values of Sconf, HEAs have several other special properties. These include the effect of sluggish (slow) diffusion [8], which was previously considered to be one of the key HEA characteristics. However, the recently published results of diffusion measurements by the method of radioactive isotopes in HEAs [9,10,11,12] do not confirm the hypothesis of slow diffusion in these alloys. Therefore, this effect requires verification and special studies.

When choosing the chemical composition of heat-resistant HEAs, it should be borne in mind that most refractory metals have a BCC lattice, which is characterized by a tendency to brittle fracture [13]. Considering this circumstance, the authors of [14] proposed using an equiatomic alloy (Ti, Zr, Hf, Ta, Nb, Mo) with the matrix (β-phase) of a promising HEA with a BCC lattice. This alloy has acceptable ductility; however, at room temperature, in addition to the β-phase, a certain amount of the second phase is observed in it. Therefore, to carry out diffusion measurements, it is necessary to determine beforehand the temperature range of β-phase stability, in which it is advisable to carry out diffusion annealing.

Powder metallurgy in combination with hot isostatic pressing (HIP) is supposed to be used for the production of alloys. This technology makes it possible to obtain objects with homogeneous structures and chemical compositions [15]. To determine the temperature range of the stability of the β-phase in the alloy under consideration, the results of the calculation of the phase diagram of the alloy by Thermo-Calc software were used [16].

The purpose of this work is to study the interdiffusion of elements in a diffusion pair consisting of Ti and equiatomic high–entropy alloy (HEA) TiZrHfNbTaMo in the temperature range of 1473–1673 K.

## 2. Materials and Methods

The refractory equiatomic alloy TiZrHfNbTaMo (HEA) was obtained by the mechanochemical synthesis method of elementary refractory metal powder mixtures in a planetary mill, followed by alloy powders compaction via HIP. Details of the methodology for obtaining compact samples of the alloy are described in [15].

The starting elementary powder mixtures for the research were prepared using extra-pure-grade (99.96%) Ti, Zr, Hf, Mo, Nb, and Ta powders 40 to 80 μm in size of the particle. The powder mixtures were ground in a Fritsch (P-7) planetary mill under an Ar atmosphere at a powder-to-ball weight ratio of 1:8. Grinding was continued as long as their X-ray diffraction patterns indicated changes in the diffraction peak shape and/or in the composition of the phase. HEA powders were compacted in stainless steel capsuled that were pumped down and electron beam sealed. HIP was carried out at a temperature of 1473 K and a holding time of at least two hours.

The diffusion experiment consisted of five stages:(1)Obtaining HEA/Ti diffusion pairs by pre-welding the HEA samples along with titanium plates and quality control of the welded seams;(2)Diffusion annealing at different temperatures during the experiment time;(3)Diffusion penetration zone identification in the diffusion pair after annealing by a mechanical treatment aid using grinding abrasive paper and felt disks with a concentrated colloidal silicon suspension;(4)Interdiffusion zone profiling using EPMA;(5)Analytical processing of the obtained diffusion profiles, calculation of HEA diffusion characteristics by the Hall’s method, and discussion of the results obtained.

The goal of the first stage was to obtain diffusion pairs. For this purpose, a low-temperature welding technique of titanium plates and HEA cylindrical samples was developed [17]. Titanium plates with a thickness of ~5 mm, a length of ~2 mm, and a width of ~6 mm were used; the diameter of the HEA samples was ~5 mm and the height was ~3 mm.

Before pre-welding, the samples were ground and polished on grinding equipment via grinding abrasive paper, after which the samples were polished by a felt disk applied with a concentrated colloidal silicon suspension (the silicon particles size in the suspension ~0.05 µm) to a mirror finish. After these operations, the samples were washed in an ultrasonic bath. For pre-welding, cleared titanium and HEA samples were clamped into laboratory stainless steel Hoffmann clamps between tantalum substrates on the clamp sponges. The pressure between the clamp sponges after clamping the samples was ~2 MPa. The clamp with clamped samples was placed in an Ar atmosphere in a quartz reactor. The reactor was installed in a vertical tubular furnace, where the samples were welded by means of thermocyclic treatment. The mode of diffusion pairs preparation consisted of thermal cycling near a Ti polymorphic transformation temperature. One cycle consisted of heating the samples to a temperature of 1152 K and cooling them down to room temperature via extracting the quartz reactor with the researched samples from the furnace without maintaining a temperature for that sample during the experiment time. A thermocouple of TVR A1 was used to determine the change in temperature of the samples in the quartz reactor. After welding, the resulting welded diffusion pairs were removed from the clamps, washed in an ultrasonic bath, and then treated with acetone. For weld quality control, the diffusion pair cross-section was polished until a clear zone of the welded joint was identified.

The study of welded joints after welding and annealing was carried out on a JEOL JSM-6480LV low vacuum scanning electron microscope (SEM) from JEOL (Tokyo, Japan); with an INCA ENERGY DryCool energy-dispersive spectrometry attachment by Oxford Instruments (UK), beside the NavaNanoSEM microscope by FEI Company (USA) with a Bruker attachment (Germany). 

The concentration profiles of the researched elements were obtained using above equipment. The operating voltage was 15 kV, the start diameter of the electron beam was about 100 nm, and the vacuum within the chamber was 3 × 10^−3^ Pa. The analysis region diameter was approximately 2–3 μm. The instrument tool error did not exceed ~0.6 of a percent.

Annealing of the researched samples was carried out in a vacuum electric furnace of resistance in vacuum 6.65 × 10^−3^ Pa.

Thermodynamic calculations were carried out using Thermo-Calc software v. 2021b and the RCCA database constructed by the authors.

## 3. Experimental Results

Figure 1a shows the SEM image of the welded seam between the titanium samples (darker area) and the HEA sample (lighter area) after pre-welding. The welded seam turned out to be solid and there were no pores and shells in the joint area.

Using EPMA, a narrow HEA elements interdiffusion zone was detected in the diffusion pair, which did not exceed a few microns. Figure 1b demonstrates the dependence graph X-ray energy of each HEAs’ alloying system element on the depth penetration obtained using EPMA by crosswise scanning the welded seam after welding of the investigated samples.

In the second stage of the experiment, the obtained diffusion pairs were annealed at temperatures of 1473, 1573, and 1673 K for 12, 9, and 6 h, respectively. The annealing temperatures were selected according to BCC solid solution existence region of the HEA on the property diagram (Figure 2).

The multicomponent β solid solution exists in the region, which one can see in the phase diagram (Figure 2). It lies in the temperature range of 1473–2150 K, therefore, accordingly, this range of diffusion annealing temperatures was chosen.

In the third stage, after annealing, the diffusion pairs were removed from the furnace and machined until interdiffusion zones were identified.

In the fourth stage of the experiment, profiling was carried out using EPMA in the direction of the diffusion flow. In Figure 3, typical elemental EDS maps near the welded seam are shown. A typical view of the polished surface is shown in Figure 4. The lower, darker part corresponds to Ti and more light part corresponds to the alloy.

The places of chemical analysis by EPMA are shown by points. The typical spectra are shown in Figure 5.

Figure 6 shows typical data on the dependence of spectrum lines intensity. It changes for the majority elements of multicomponent alloy at a distance of 150–200 μm. The sharp fall in signal intensity happens at a depth of about 50 μm.

In Table 1, the data are shown, recalculated according to the line intensity, and the concentrations of the elements in the diffusion zone that correspond to Figure 4.

The concentration values obtained by the EPMA method were put onto an electronic scale coordinate grid in a Kompas 3D computer-aided design program. Figure 7 shows the diffusion profile of Hf after annealing at 1473 K for 12 h obtained by EPMA. Most parts of the diffusion profiles look similar for the other system elements.

Other diffusion profiles of the researched metal systems are given in Appendix A.

## 4. Calculation of Diffusion Parameters

The primary analysis shows that one can see the difference between lines Zr and Hf, but for all other elements, the concentration profile is very close.

A characteristic feature of this dependence, as well as for all elements of THE researched system, is its obviously asymmetric form, corresponding to the diffusion coefficient (*D*) dependence on the concentration. Such dependencies are most often described by the Matano–Boltzmann method [18]. Therefore, the Matano plane was determined for all of the obtained concentration curves; the Hf curve is shown in Figure 7. For this case, the Matano plane position is 295 microns from the coordinate zero. Note that the zero coordinate in the further calculations is counted from the Matano plane since we are interested in the interdiffusion of system elements in the HEA’s area.

Although in general the Matano plane does not coincide with the interface, the part of the concentration curve to the left side relative to the Matano plane corresponds to element diffusion in Ti and for the right side, it is according to the element diffusion in HEA region (Figure 4).

The fifth experimental stage included experimental data analytical processing obtained in the previous stage and calculation of the diffusion parameters. The Matano method is essentially graphical, and the error in its application grows at the concentration curve ends. To determine the diffusion coefficients in the concentration region between the concentration curve ends and the Matano plane, although closer to the ends, a smaller error is given by Hall’s method [19], based on an experimental concentration curve analytical approximation in that part of it, where the Matano method leads to significant errors. For determining the diffusion coefficients by Hall’s method, the experimental concentration curve *c*(*x*) for all elements of the studied system was plotted in relative coordinates: *c*/*c*_2_ (*c*_2_ equal 16.7 at.% for all elements of six component equiatomic HEAs except Ti; *c*_2_ of titanium profile ≈ 100 at.%) as distance function *f*(*x*) or *f*(λ) where λ = *x*/*t*^1/2^ (here *t* is the annealing time) on the probability diagram; the *U*-value was found for each experimental point of this function accordingly the equation:(1)$$\frac{c-c_{1}}{c_{2}-c_{1}}=\frac{1}{2}\left[1+\mathrm{erf}\left(U\right)\right]$$

Here *c*_1_ is the lowest element concentration of each element diffusion profile.

Next, a curve *U*(λ) was constructed, and a rectilinear section was found on this curve between the Matano plane and a horizontal line passing through *c*_2_ = 16.7 at.%. The value was calculated from the Matano plane, i.e., from λ = 0.

The probability diagram *U*(λ) for the Hf concentration profile, obtained after annealing of the studied diffusion pair at 1473 K, is presented in Figure 8, and the Matano plane is matched on the ordinate axis here.

Figure 8 shows the line segment built using a linear fit of the dependence *U*(λ) illustrated in the alloy region via the red color. To calculate the diffusion coefficient *D* of the studied system, the coefficients h and k found from Equation (2) are used:*U* = *h*λ + *k*(2)

For the Hf case (experimental data received after annealing at 1473 K) equation of the line accords *U* = 4.8λ − 0.11 (Figure 8). The coefficients h and k are used for further calculations using Hall’s method [19,20,21] via Expression (3):
(3)$$D\left(c\right)=\frac{1}{4h^{2}}\left\{1+k\sqrt{\pi }\exp\left(u^{2}\right)\left[1+\mathrm{erf}\left(u\right)\right]\right\}$$
The interdiffusion coefficients of all of the researched elements for the HEA’s zone obtained at diffusion annealing temperatures of 1473, 1573, and 1673 K for 12, 9, and 6 h, respectively, are presented in Table 2. Concerning the *D*(*c*) function, it draws attention to the weak dependence of the diffusion coefficients on the concentration in the alloy region for all of the metallic concentration profiles cases. To confirm this one, for instance, in Figure 8, Hf diffusion coefficients can be calculated using dots from the red line segment so the first dot coordinates (0.42 and 0.11) and the second dot coordinates (1.2 and 0.27) can be taken on this graph for Hall’s method calculation. The *D* of the first dot is 9.5 × 10^−15^ m^2^/s and the *D* of second dot is equal to 1.0 × 10^−14^ m^2^/s. In addition, some dots of the line segment equal to *D* = 9.8 × 10^−15^ m^2^/s. For all other diffusion profiles of the HEA’s alloying system elements, calculations were carried out in the same way.

Calculated diffusion characteristics values, such as activation energies Q and pre-exponential factors *D*_0_, are shown in Table 2 and demonstrate the HEA’s elements behavior in the alloy diffusion zone of the HEA/Ti diffusion pair.

The interdiffusion coefficients values for all of the researched system elements of the diffusion pair in the alloy area are demonstrated on the Arrhenius diagram (Figure 9).

## 5. Discussion

The interdiffusion in the paired equiatomic alloy TiZrHfNbTaMo and Ti (BCC lattice, β-phase) was investigated. The diffusion pairs were annealed at temperatures of 1473, 1573, and 1673 K for 12, 9, and 6 h, respectively. The concentration values of all six alloying elements of the HEAs were obtained by the EPMA method. All of the obtained diffusion penetration profiles were asymmetrical, which indicates that the diffusion in the titanium is faster than in the alloy. This is not surprising, since the melting point of titanium is lower than that of other elements.

In this investigation stage, the most interesting are interdiffusion parameters of all six elements in the alloy. These parameters (interdiffusion coefficients, activation energies, and pre-exponential factors) determined by Hall’s method are shown in Table 2 and Table 3, and in Figure 9.

Note that although the melting points of all six elements, except for Ti and Zr, differ from 3290 K for Ta to 2500 K for Hf and the self-diffusion activation energies differ by more than 400 kJ/mol for Ta, Nb, and Mo and more than 300 kJ/mol for Hf, the interdiffusion coefficients and activation energies are much closer to each other. Thus, the activation energies differ by only 10%, from 294 kJ/mol for Ta to 272 kJ/mol for Hf. This does not contradict Darken’s analysis of the Kirkendol effect.

In Table 4, the values of self-diffusion activation energies for each element of the HEA from the literature are given for comparison.

The exception is the interdiffusion parameters of Ti and Zr in the HEA, which differ from the others by more than two times (139 kJ/mol for Ti and 126 kJ/mol for Zr). A similar effect was observed in a recently published paper [37], in which the diffusion of Zr in an HEA of the same composition was researched by the isotopic method, and activation energies from 134 to 172 kJ/mol were obtained. Such results remain unexplained. However, the most probable explanation is connected with the difference in mechanism. Ta, Nb, Mo, and Hf diffuse by the vacancy mechanism, while Zr and Ti diffuse by the interstitial mechanism, the much faster one [38].

Two other reasons can be assumed. The first is the relatively low melting temperatures of Ti and Zr and the second is the fact that both elements experienced polymorphic transformation during cooling, which can affect the measured results.

The last remark concerns the sluggish diffusion effect in HEAs. It may be of interest to compare the experimental data obtained with the parameters of self-diffusion. Table 3 shows the data of the literary sources on the self-diffusion of these elements. It should be noted that there is a large spread of data. However, despite this, the activation energy of self-diffusion for more cases is greater than the activation energy of interdiffusion, which is not favorable to the sluggish effect.

## 6. Conclusions

The interdiffusion process in the β-phase BCC equiatomic high entropy alloy (HEA) TiZrHfNbTaMo was investigated after annealing at temperatures of 1473, 1573, and 1673 K for 12, 9, and 6 h, respectively. The interdiffusion parameters of all six elements in the alloy were determined by the Hall method. All of the obtained diffusion penetration profiles are asymmetric which indicates that diffusion in titanium is faster than in the alloy. According to our analysis, the interdiffusion parameters of the components in the multicomponent alloy are close to one another if their diffusion mechanism is the same.

The most probable explanation for the difference between interdiffusion parameters Ti and Zr and the others is connected to a difference in the mechanism. In the investigated system, Mo, Nb, Ta, and Hf diffuse by the vacancy mechanism, but Ti and Zr diffuse by the interstitialcy mechanism much more rapidly.

The activation energy of self-diffusion is greater than the activation energy of interdiffusion, which is not favorable to the sluggish effect for the researched elements of the system.

## Figures and Tables

**Figure 1 entropy-25-00490-f001:**
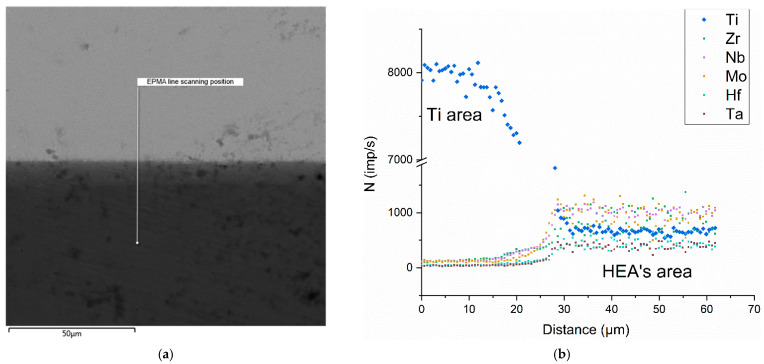
Typical welded seam after thermocyclic treatment and EPMA line scanning position across the HEA/Ti pair interface (**a**) and the results of EPMA scanning (**b**) after pre-welding.

**Figure 2 entropy-25-00490-f002:**
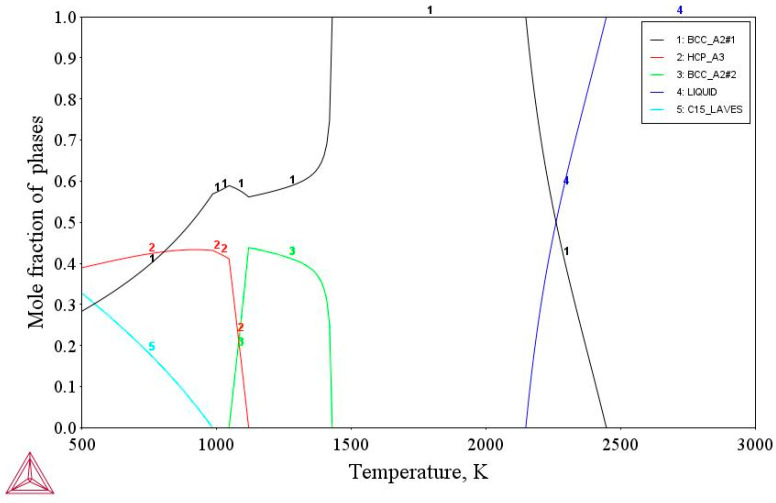
HEAs’ calculated property diagram.

**Figure 3 entropy-25-00490-f003:**
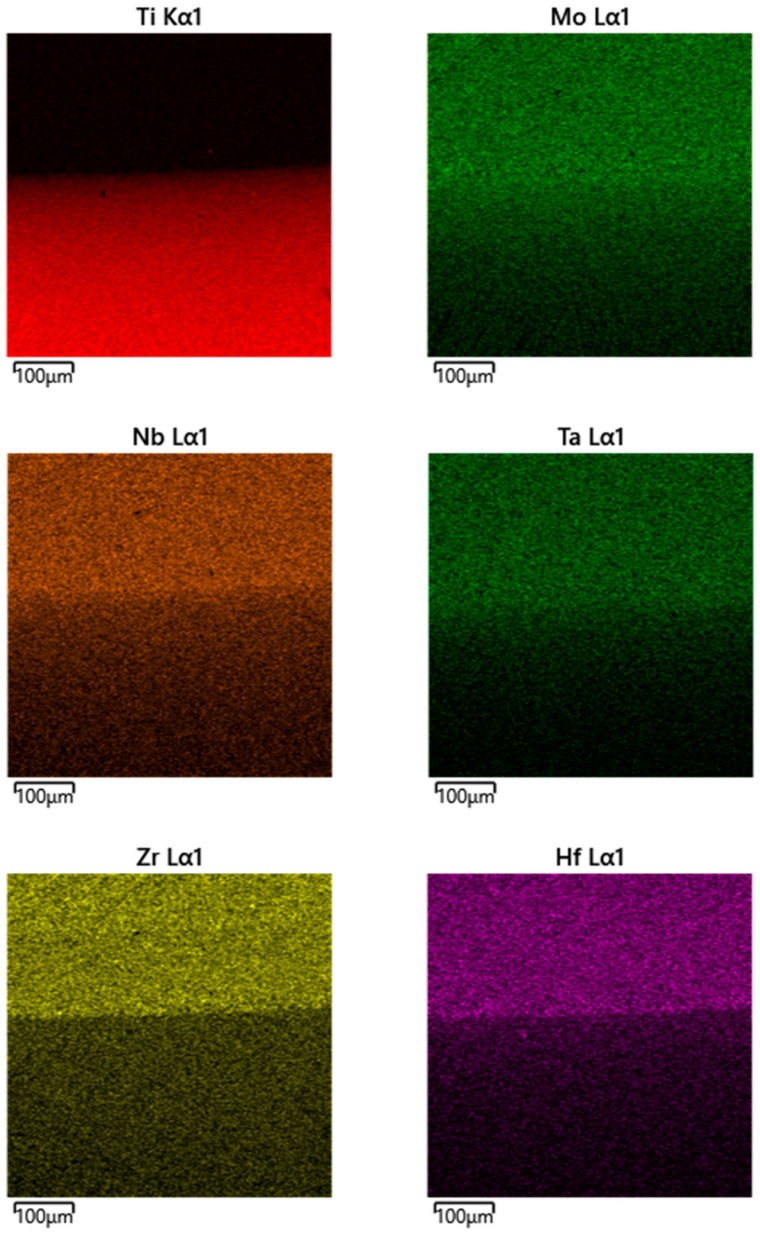
EDS elemental maps showing elemental distribution in the interdiffusion zone after the HEA/Ti pair’s annealing at 1473 K for 12 h.

**Figure 4 entropy-25-00490-f004:**
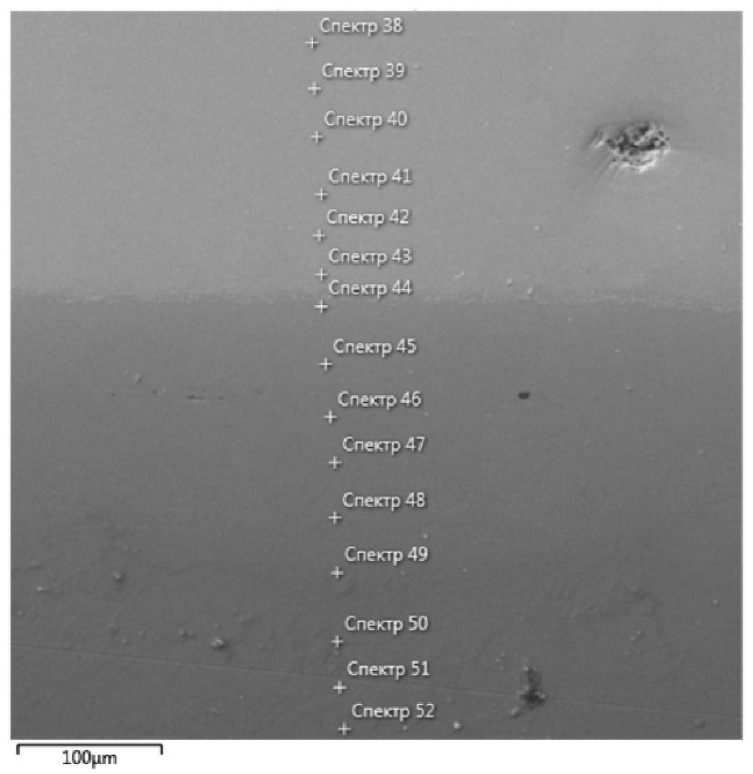
SEM image of diffusion area after the HEA/Ti pair’s annealing at 1473 K for 12 h (the white crosses are in accordance with each spectra number from 38 to 52 and the metal concentration values in Table 1 accordingly).

**Figure 5 entropy-25-00490-f005:**
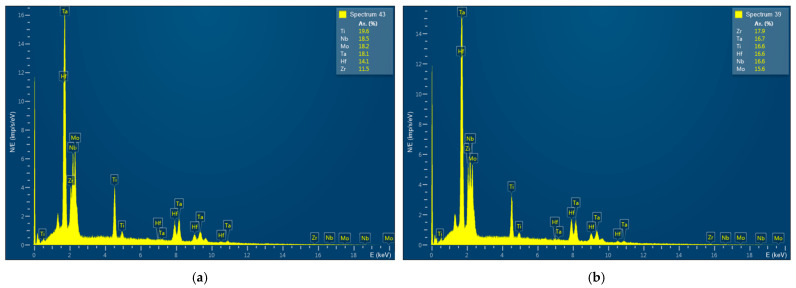
Typical spectra (that corresponds to Figure 4) near the interface (**a**); in the alloy region (**b**).

**Figure 6 entropy-25-00490-f006:**
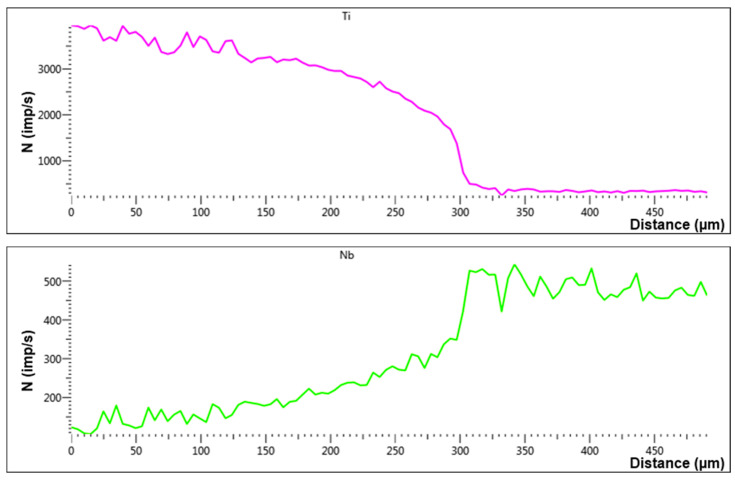
Intensity profiles of Ti and Nb in the interdiffusion zone of the researched system after the HEA/Ti pair’s annealing at 1473 K for 12 h.

**Figure 7 entropy-25-00490-f007:**
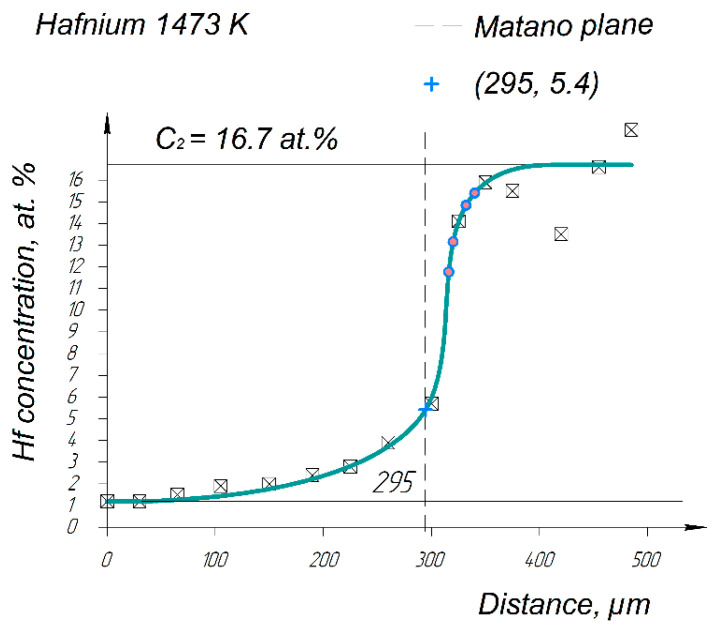
Hf concentration profile after the HEA/Ti pair’s annealing at 1473 K for 12 h and the Matano plane (dashed line) corresponding to the coordinate of the 295 microns in the graph. The Ti zone on the left side of the Matano plane and the HEA zone on the right side. (Pink points are used in diffusion coefficients calculation via Hall's method).

**Figure 8 entropy-25-00490-f008:**
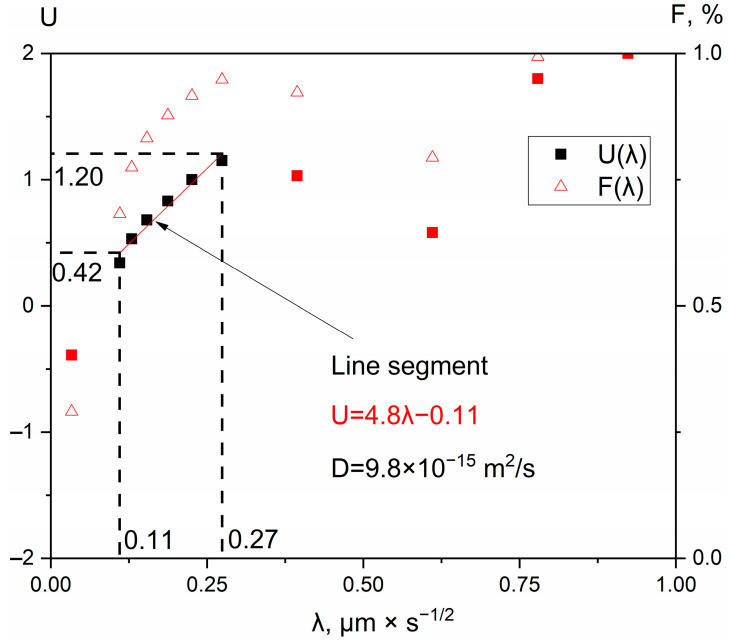
Probability plot of F = (*c* − *c*_1_)/(*c*_2_ − *c*_1_) vs. λ for the HEA/Ti diffusion pair corresponding to the experimental data obtained after the HEA/Ti pair’s annealing at 1473 K for 12 h: *c*_2_ = 16.7 at.%; *c*_1_ = 1.2 at.%; *c* = *c*_Hf_; *t* = 43,200 s. (Red symbols did not participate in the linear approximation).

**Figure 9 entropy-25-00490-f009:**
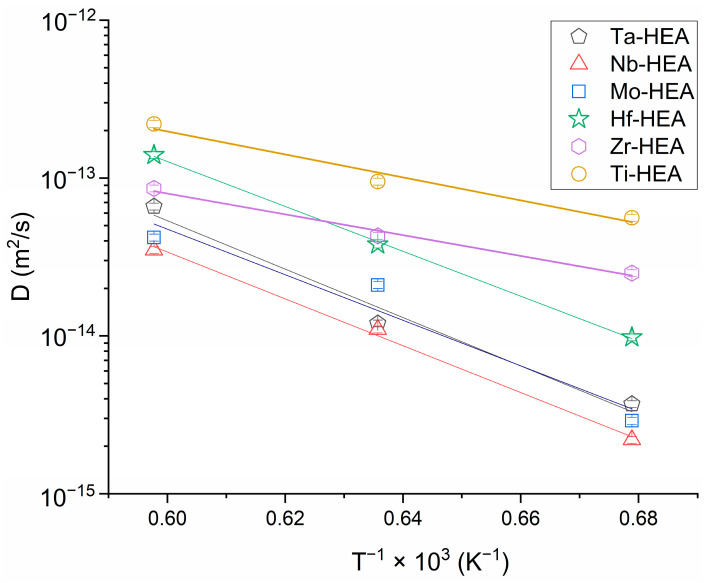
Arrhenius plot for HEA elements of the diffusion pair according to the produced experimental data.

**Table 1 entropy-25-00490-t001:** Concentrations of the elements in the diffusion zone.

Spectrum Number	Distance, µm	Ti	Zr	Hf	Mo	Ta	Nb
		at.%					
38	0	15.6	20.6	18.3	14.7	14.7	14.7
39	35	16.6	17.9	16.6	15.6	16.7	16.6
40	70	17.6	9.3	13.5	19.7	19.6	19.4
41	105	17.0	13.4	15.5	18.4	18.4	17.3
42	140	17.8	14.3	15.9	17.3	16.7	18
43	175	19.6	11.5	14.1	18.2	18.1	18.5
44	210	60.4	3.6	5.7	9.8	10	8.4
45	245	75.6	2.9	3.9	5.2	5.1	4.6
46	280	80	2.7	2.7	3.5	3.5	3.3
47	315	83.3	2.5	2.4	2.8	2.7	3
48	350	85.5	2.1	2	2.1	2	2.3
49	385	87.5	2.1	1.9	1.6	1.4	1.7
50	420	87.3	2.2	1.5	1.6	1.1	1.7

**Table 2 entropy-25-00490-t002:** HEA’s alloying system elements diffusion coefficients that represent the zone of diffusion in the alloy area (the relative error of determining the diffusion coefficients is in limits ± 5%).

Annealing Temperature/Diffusion Coefficient of the HEA’s Element	*D* (m^2^/s)					
	Hf	Nb	Ta	Mo	Zr	Ti
1673 K	1.4 × 10^−13^	3.5 × 10^−14^	6.6 × 10^−14^	4.2 × 10^−14^	8.6 × 10^−14^	2.2 × 10^−13^
1573 K	3.8 × 10^−14^	1.1 × 10^−14^	1.2 × 10^−14^	2.1 × 10^−14^	4.3 × 10^−14^	9.5 × 10^−14^
1473 K	9.8 × 10^−15^	2.2 × 10^−15^	3.7 × 10^−15^	2.9 × 10^−15^	2.5 × 10^−14^	5.6 × 10^−14^

**Table 3 entropy-25-00490-t003:** Activation energies and pre-exponential factors of all of the HEA’s elements in the diffusion area in the alloy region (the relative error in determining the diffusion characteristics is ± 6%).

Diffusion Characteristics/HEA’s Element	Hf	Nb	Ta	Mo	Zr	Ti
*Q* (kJ/mol)	272	284	294	276	126	139
*D*_0_ (m^2^/s)	4.3 × 10^−5^	2.710^−5^	8.7 × 10^−5^	2.1 × 10^−5^	7.1 × 10^−10^	4.6 × 10^−9^

**Table 4 entropy-25-00490-t004:** Activation energies of self-diffusion of some elements (Q_self_) accordingly the alloying system of the HEA from literary sources.

Element	Temperature Interval (K)	Q_self_ (kJ/mol)	Literary Primary Source
Hf	1173–1633	323.0	[22]
	1437–1883	348.3	[23]
Nb	1151–2668	402.0	[24]
	1421–2509	397.3	[25]
	1354–2695	438.3	[26]
	1151–2695	503.0	[27]
Ta	1523–2558	413.3	[28]
		370.8	[27]
Mo	1360–2773	488.2	[29]
	1973–2193	464.8	[30]
	2073–2448	481.5	[31]
Zr	1441–1776	126.0	[32]
	1174–2020	116.0	[33]
	1189–2000	184.4	[34]
	1174–2020	81.1	[27]
Ti	1176–1893	328.3	[35]
	1172–1813	130.6	[36]
	1172–1893	119.7	[27]

## Data Availability

Not applicable.

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
