# Peer review of "Interdiffusion in Refractory Metal System with a BCC Lattice: Ti/TiZrHfNbTaMo"

_entropy, 2023, doi:10.3390/e25030490_

Round 1
Reviewer 1 Report
I think the present research is one impressive item on refractory alloy system. There are several important points to be significantly drawn.
(1) More clear demonstration of the experimental preparation processes should be provided, and an illustrated figure might be more friendly to the readers.
(2) The raw experimental composition curves are not available. The authors should be asked to present all raw composition curves for general validation by the readers and reviewers as least as a supplementary file.
(3) A specific question is devoted to the consideration of the off-diagonal terms of the interdiffusion matrices. Comments on the availability of the off-diagonal terms should be presented.
(4) I would like to kindly refer the authors to the numerical inverse method (High-throughput determination of high-quality interdiffusion coefficients in metallic solids: A review. J. Mater. Sci. 2020, 55, 10303–10338.), which might be able to extract the composition-dependent information from the entire dataset of composition curves. May the authors leave a future prospect on such work which might potentially inspire readers of the same research interests.
(5) The discussion should be rearranged, and conclusions should be shortened.
(6) Last but not least, the sentences should be significantly reviewed, checked, and improved.
Author Response
Dear editors, Dear colleagues
Dear reviewers,
We are very grateful for professional, detailed and critical analysis of the manuscript. Especially our gratitude for the recommendation the review in J.Mat.Sci. We did not see it. Unfortunately, our possibility to read the foreign technical literature is now very restricted.
We have done many improvements and additions in our manuscript, so it was impossible to mark them all. Nevertheless, some of them.
Reviewer's comments:
I think the present research is one impressive item on refractory alloy system. There are several important points to be significantly drawn.
- More clear demonstration of the experimental preparation processes should be provided, and an illustrated figure might be more friendly to the readers.
Answer.
We have described the processes of obtaining high entropy alloy samples and preparing that samples for pre-welding and diffusion annealing in more detail.
- The raw experimental composition curves are not available. The authors should be asked to present all raw composition curves for general validation by the readers and reviewers as least as a supplementary file.
Answer.
We have added an appendix at the end of the manuscript that includes composition curves of the researched system.
- A specific question is devoted to the consideration of the off-diagonal terms of the interdiffusion matrices. Comments on the availability of the off-diagonal terms should be presented.
Answer.
It was not our goal to obtain the full dependence D(C) including off-diagonal terms — it was too hard problem. We obtained only one point between the Matano plane and Cmax. In addition we recognized that dependence D(C) is very weak, especially near C(X).
- I would like to kindly refer the authors to the numerical inverse method (High-throughput determination of high-quality interdiffusion coefficients in metallic solids: A review. J. Mater. Sci. 2020, 55, 10303–10338.), which might be able to extract the composition-dependent information from the entire dataset of composition curves. May the authors leave a future prospect on such work which might potentially inspire readers of the same research interests.
Answer.
Our gratitude for the recommendation the review in J.Mat.Sci. Unfortunately, our possibility to read the foreign technical literature is now very restricted.
- The discussion should be rearranged, and conclusions should be shortened.
Answer.
We increased introduction defining more precisely the goal of our paper and rearranged discussion widening analysis of results. We shortened conclusion. It is now more precise.
- Last but not least, the sentences should be significantly reviewed, checked, and improved.
Answer.
About language. We asked our colleague, for whom English is native, to read our manuscript. He improved many places.
That Note: corrections made are highlighted in blue in the manuscript.
Reviewer 2 Report
I have some doubts with the followings:
- the English of the article is poor. I cannot understand the basic statements, for this reason I have to use my own imagination to figure out, what is the meaning.
- Sample preparation: should be improved, the information provided is not enough for understanding. For example:
- Experiment first stage goal was to obtain diffusion pairs. For this purpose, low-temperature welding technique of titanium plates and HEA’s cylindrical samples was developed [17]. AND until a mirror reflection appeared.
- This article is dealing with diffusion, so the preparation of the samples is a critical point.
- Authors claim they used EDS analysis (or EPMA) for chemical analysis and the error of this is around 0.3 per cent. This is really good resolution, however, I have to ask some background data.
- They define the Mattano plane using the results coming from EDS analysis. If this is wrong, the whole calculation relies on this data.
Author Response
Dear editors, Dear colleagues
Dear reviewers,
We are very grateful for professional, detailed and critical analysis of the manuscript. Especially our gratitude for the recommendation the review in J.Mat.Sci. We did not see it. Unfortunately, our possibility to read the foreign technical literature is now very restricted.
We have done many improvements and additions in our manuscript, so it was impossible to mark them all. Nevertheless, some of them.
Reviewer's comments:
I have some doubts with the followings:
- The English of the article is poor. I cannot understand the basic statements, for this reason I have to use my own imagination to figure out, what is the meaning.
Answer.
We asked our colleague, for whom English is native, to read our manuscript. He improved many places.
- Sample preparation: should be improved, the information provided is not enough for understanding. For example:
Experiment first stage goal was to obtain diffusion pairs. For this purpose, low-temperature welding technique of titanium plates and HEA’s cylindrical samples was developed [17]. AND until a mirror reflection appeared.
This article is dealing with diffusion, so the preparation of the samples is a critical point.
Answer.
We have described the processes of obtaining high entropy alloy samples and preparing that samples for pre-welding and diffusion annealing in more detail.
- Authors claim they used EDS analysis (or EPMA) for chemical analysis and the error of this is around 0.3 percent. This is really good resolution, however, I have to ask some background data.
They define the Mattano plane using the results coming from EDS analysis. If this is wrong, the whole calculation relies on this data.
Answer.
We added some results to make their demonstration clearer. The part of results relates to EPMA background and make C(X) more precise. The second contain all figures C(X) for all elements and temperatures in supplement.
We have clarified the information regarding the equipment in “Materials and methods” part.
Furthermore, we have added an appendix at the end of our manuscript that includes concentration curves of the researched system metals and corresponding that Matano planes.
That Note: corrections made are highlighted in blue in the manuscript.
Reviewer 3 Report
Dear authors,
Thank you for the article on diffusion in
high entropy alloys. I have some comments on improving the
quality.
1. The introduction is very short and the state of
research is not sufficiently elaborated, please rework clearly here.
2. The objective is too strongly focused. Please
give a hint where the system under consideration and your results can be applied.
3. The presentation of the HIP and welding process
is unfortunately almost completely missing. Process variables and methodology should be presented.
4. The description of the heat treatment is unfortunately
too short, please make it more detailed.
5. The description of the results is far too
short. Please be much more detailed
6. The conclusions are much too long and contain
elements of the discussion. Please split into discussion and conclusions.
The discussion should be detailed.
Author Response
Dear editors, Dear colleagues
Dear reviewers,
We are very grateful for professional, detailed and critical analysis of the manuscript. Especially our gratitude for the recommendation the review in J.Mat.Sci. We did not see it. Unfortunately, our possibility to read the foreign technical literature is now very restricted.
We have done many improvements and additions in our manuscript, so it was impossible to mark them all. Nevertheless, some of them.
Reviewer's comments:
Dear authors,
Thank you for the article on diffusion in high entropy alloys. I have some comments on improving the quality.
- The introduction is very short and the state of research is not sufficiently elaborated, please rework clearly here.
- The objective is too strongly focused. Please give a hint where the system under consideration and your results can be applied.
Answer.
We increased introduction defining more precisely the goal of our paper.
- The presentation of the HIP and welding process is unfortunately almost completely missing. Process variables and methodology should be presented.
- The description of the heat treatment is unfortunately too short, please make it more detailed.
Answer.
We have enlarged section «Materials and Methods».
We have described the processes of obtaining high entropy alloy samples via HIP and preparing that samples for pre-welding and diffusion annealing in more detail.
- The description of the results is far too
short. Please be much more detailed.
Answer.
We rearranged discussion widening analysis of results.
- The conclusions are much too long and contain
elements of the discussion. Please split into discussion and conclusions.
The discussion should be detailed.
Answer.
We shortened conclusion. it is now more precise.
That Note: corrections made are highlighted in blue in the manuscript.
Round 2
Reviewer 1 Report
The effort from the authors is greatly appreciated, thought I personally considered that the presentation of the results can be significantly improved. With Figure A1-A17, I would like to propose for accepting the present work, though there are a few questions to be significantly drawn.
1. The explicit relation between D and U might be provided for the goodness of audients.
2. Reasonable explanation of the change of terminal composition of Ti in Fig. A1 should be provided. That is, leave some comments on the reason why C2 deviates from pure Ti. Moreover, the authors are encouraged to double-check this by revealing composition profiles for 1473K at different annealing times other than 9 hrs.
3. There are too many short sentences, and some of them should be rearranged.
Author Response
Dear colleagues, We are grateful for the second reviews. Three answers on the questions in the review # 1. 1. The relation between D and U is provided (see in the text). 2. Of coarse, the reviewer is absolutely right. We were obliged to receive 100% for a pure Ti instead of 85% as at Fig.1. Unfortunately, we could not do it because the sample was spoiled. It was the last and we had not the other one. To receive the new one requires too much time; We hope to do it in the future . 3. We rearranged the short sentences which are not necessary (the improvements are marked in the text).Reviewer 2 Report
The quality of the manuscript has improved a lot, in favor of comprehensibility. Thank you for your work and keep pushing science.
Author Response
Dear colleagues, We are grateful for the second reviews.
Very sincerely,
Mikhail Razumovsky, Boris Bokstein
Reviewer 3 Report
Thank you very much for the comments, i have no further requests
Author Response

(The authors gave the same response as above.)
